# Phage Display-Based Homing Peptide-Daunomycin Conjugates for Selective Drug Targeting to PANC-1 Pancreatic Cancer

**DOI:** 10.3390/pharmaceutics12060576

**Published:** 2020-06-22

**Authors:** Levente E. Dókus, Eszter Lajkó, Ivan Ranđelović, Diána Mező, Gitta Schlosser, László Kőhidai, József Tóvári, Gábor Mező

**Affiliations:** 1MTA-ELTE Research Group of Peptide Chemistry, Hungarian Academy of Sciences, 1117 Budapest, Hungary; doehabt@caesar.elte.hu (L.E.D.); schlosser@caesar.elte.hu (G.S.); 2Institute of Chemistry, Faculty of Sciences, Eötvös Loránd University, 1117 Budapest, Hungary; 3Department of Genetics, Cell- and Immunobiology, Semmelweis University, 1089 Budapest, Hungary; lajko.eszter@med.semmelweis-univ.hu (E.L.); mezo.diana@med.semmelweis-univ.hu (D.M.); kohidai.laszlo@med.semmelweis-univ.hu (L.K.); 4National Institute of Oncology, Department of Experimental Pharmacology, 1122 Budapest, Hungary; ivan.randel@gmail.com (I.R.); tozsi@oncol.hu (J.T.)

**Keywords:** pancreatic cancer, targeted tumor therapy, homing peptide, antitumor peptide conjugates, daunomycin, oxime linkage

## Abstract

The Pancreatic Ductal Adenocarcinoma (PDAC) is one of the most aggressive and dangerous cancerous diseases, leading to a high rate of mortality. Therefore, the development of new, more efficient treatment approaches is necessary to cure this illness. Peptide-based drug targeting provides a new tool for this purpose. Previously, a hexapeptide Cys-Lys-Ala-Ala-Lys-Asn (CKAAKN) was applied efficiently as the homing device for drug-loaded nanostructures in PDAC cells. In this research, Cys was replaced by Ser in the sequence and this new SKAAKN targeting moiety was used in conjugates containing daunomycin (Dau). Five different structures were developed and tested. The results indicated that linear versions with one Dau were not effective on PANC-1 cells in vitro; however, branched conjugates with two Dau molecules showed significant antitumor activity. Differences in the antitumor effect of the conjugates could be explained with the different cellular uptake and lysosomal degradation. The most efficient conjugate was *Dau=Aoa*-GFLG-K(*Dau=Aoa*)SKAAKN-*OH* (conjugate **4**) that also showed significant tumor growth inhibition on *s.c.* implanted PANC-1 tumor-bearing mice with negligible side effects. Our novel results suggest that peptide-based drug delivery systems could be a promising tool for the treatment of pancreatic cancers.

## 1. Introduction

Pancreatic Ductal Adenocarcinoma (PDAC) is one of the most aggressive and dangerous cancerous diseases with a high mortality rate [1]. In the USA, more than 55,000 new cases were estimated in 2018 which is 3–4% of the all newly diagnosed cancer cases. Approximately 80% of these incidences will lead to death within a year [2]. The average 5-year survival rate is less than 5% [3]. The main reason for the high mortality of pancreatic cancer patients could be very poor prognosis. The early diagnosis of PDAC is still difficult, and most patients have already progressed to not operable and incurable statuses at the recognition of the disease [4]. In addition, the chemotherapy applied to treat pancreatic cancers is usually ineffective due to the fast development of resistance. Chemotherapy causes many side effects because of the low selectivity of the currently used drugs [5]. Furthermore, the hypovascularity of PDAC and a dense desmoplastic stroma that create barriers restrict drug delivery to the tumor site [6]. Therefore, the design of efficient anticancer agents against PADC is one of the most challenging tasks for scientists working on cancer research [7]. Targeted tumor therapy could be a promising strategy to overcome these drawbacks in pancreatic cancer treatment—similar to other types of cancers [8]. Targeted tumor therapy is based on targeting tumor-specific or overexpressed receptors or other cell surface compartments on tumor cells that can be recognized selectively by antibodies or small molecules like folic acid or peptides [9,10]. Drug molecules attached to these homing moieties can enter specifically into tumor cells, resulting in selective toxicity without causing toxic side effects in healthy tissues. The application of small molecule drug conjugates (SMDCs) over antibody-drug conjugate (ADCs) may have an advantage in the treatment of PDAC because SMDCs have higher tissue permeability [11].

Several homing peptides that recognize pancreatic cancer cells and could be used for drug targeting directly or as a part of nanoparticles have been described in the literature [12,13,14,15]. One of them is the CKAAKNK oligopeptide that was selected by phage display technique and which can specifically bind to tumor vessels in RIP-Tag2 transgenic mice, a prototypical mouse model of multistage pancreatic islet cell carcinoma [16]. Valetti et al. attached the CKAAKN homing peptide to a squalene (SQ) molecule via thiol-maleimide Michael addition coupling [17]. The conjugate was co-nanoprecipitated with the squalenoyl prodrug of gemcitabine (SQdFdC) resulting in nanoparticles. The construct was tested on MIA PaCa-2 human pancreatic adenocarcinoma cells, which overexpress frizzled-5 (FZD-5) receptors compared to NIH/3T3 fibroblasts. It was indicated that these cells selectively took up the nanoparticles decorated with the homing peptide by a receptor-mediated way [18]. Frizzled receptors as Wnt binding 7TM GPCRs are key players in the Wnt/β-catenin signal pathway that is commonly hyperactivated in pancreatic cancers, leading to enhanced cell proliferation [19]. In the presence of appropriate ligands, the FZD-5 receptor can be internalized, usually in a heterodimeric form [20]. Therefore, this protein is a promising target for drug targeting to tumor cells. In addition, these data indicated the efficacy of the CKAAKN peptide as a homing device—related to the Wnt-2 sequence—for targeted tumor therapy.

In our research, several SMDCs were developed, derived from the CKAAKN oligopeptide. The structure–activity relationship was investigated as well. In this work, cysteine was replaced by serine to remove the unnecessary thiol group at the conjugation site. This exchange is widely used to eliminate reactive thiol group when it is not essential for the biological activity. In addition, the substitution of Cys by Ser improves the hydrophilicity and solubility of the peptide and its conjugates. Daunomycin (Dau), as an anticancer agent was attached to the homing peptides via oxime linkage, which shows proper stability in the circulation and allows the release of an active metabolite in lysosomes [21,22]. This active metabolite contains an amino acid (Aaa) to which daunomycin is connected through an aminooxyacetyl moiety (*Dau=Aoa*-Aaa-OH). This metabolite was proved to bind to DNA; however, the binding efficacy highly depends on the type of the amino acid [22].

The PANC-1 cell line, originally derived from head pancreatic carcinoma, was applied in our studies, and has an invasive phenotype and the ability to give metastasis to the peripancreatic lymph node; thus, this cell line can be considered as an in vitro model of lymph-node-positive PDAC. The invasiveness and metastatic potential of pancreatic cancer cells has been shown to be influenced by the Wnt/β-catenin pathway. This Wnt/β-catenin pathway has also been reported as a central element of immune-escape mechanisms of pancreatic tumors by providing an environment with immune-tolerogenic cytokine and chemokine [23]. Moreover, the expression level of β-catenin, a key protein of the Wnt pathway, has been found to be well-correlated with the gemcitabine-resistance of different pancreatic cell lines, including PANC-1 cells [24]. These three characteristics of pancreatic tumor cells like PANC-1 seem to be interrelated and orchestrated by the Wnt/β-catenin pathway.

The influence of the number of drugs as well as the presence of an enzyme cleavable spacer on antitumor activity was studied in PANC-1 pancreatic cancer cells. The antitumor effect is influenced by several cellular factors; therefore, the binding, the cellular uptake and the metabolism of the conjugates were also investigated. The best compounds identified in the in vitro studies were applied in vivo experiment using subcutan (*s.c.*) developed PANC-1 tumor-bearing SCID mice. The results were compared with free drug administration.

## 2. Materials and Methods

### 2.1. Chemicals

All amino acid derivatives and Wang resin were purchased from Iris Biotech GmbH (Marktredwitz, Germany), whereas reagents for coupling and cleavage (*N,N′*-diisopropylcarbodiimide (DIC), 1,8-diazabicyclo[5.4.0]undec-7-ene (DBU), 1-hydroxybenzotriazole hydrate (HOBt), 4-(dimethylamino)pyridine (DMAP), triisopropylsilane (TIS), trifluoroacetic acid (TFA)) and ninhydrin were delivered by Sigma-Aldrich Kft. (Budapest, Hungary). Aminooxyacetic acid (Aoa) and methoxyamine were TCI (Tokyo, Japan) products. The solvents (dichloromethane (DCM), *N,N*-dimethylformamide (DMF), acetonitrile (CH_3_CN)) for synthesis and purification were obtained from Reanal (Budapest, Hungary) or VWR International Kft. (Debrecen, Hungary). Daunomycin (Dau) was donated from IVAX (Budapest, Hungary).

### 2.2. Synthesis of Peptides

Peptides were synthesized manually by solid-phase peptide synthesis on Wang resin (0.25 g, 0.52 mmol/g) using the standard protocol of Fmoc/*^t^*Bu strategy. The first amino acid derivative (5 equivalent to the resin capacity) was attached to the resin with a DIC coupling agent in the presence of 0.5 equivalent DMAP in DMF. The Fmoc group was removed with 2% DBU and 2% piperidine in DMF (four times; 2, 2, 5, 10 min, respectively). For the coupling of the following amino acid derivatives, DIC-HOBt mixture (3 equivalent each) were applied in DMF for 60 min. The *ε*-amino group of lysine used for the development of a branch was protected with 4-methyltrityl (Mtt) group that could be removed selectively next to the *tert*-butyl type protecting groups with 2% TFA and 2% TIS in DCM (7 times; 1, 1, 3, 3, 5, 10, 30 min, respectively). The aminooxyacetic acid used for the development of an oxime linkage was incorporated in its isopropylidene protected form [25] either to the *N*-terminus or to both *N*-termini (backbone and branch) of the peptides. DIC and HOBt coupling agents were used for this purpose, similar to the coupling of amino acid derivatives. The peptides were removed from the resin by cleavage with 5 mL TFA, containing 0.125 mL distilled water and 0.125 mL TIS (as scavengers). The crude product was precipitated by dry diethyl ether, dissolved in 10% acetic acid, freeze-dried and purified by RP-HPLC (Gradient I.).

### 2.3. Synthesis of Daunomycin Conjugates

In the first step, the isopropylidene protecting group was removed from the aminooxyacetyl moiety of the purified peptide derivatives by methoxyamine (in 1.5 M concentration) in 0.2 M NH_4_OAc buffer solution (pH = 5) at RT for 2 h. The reaction took place quantitatively. The unprotected products were isolated by RP-HPLC (Gradient I.). Prior to the conjugation, the solvent was evaporated and then the rest was re-dissolved in 0.2 M NH_4_OAc buffer solution (pH = 5) and 2 equivalent Dau to the peptide was added to the mixture. The reaction was carried out overnight. The reaction mixture was injected directly to the HPLC in all cases and the conjugates were separated from the excess of Dau by RP-HPLC (Gradient II.)

### 2.4. Reverse Phase High-Performance Liquid Chromatography (RP-HPLC)

The purification of the crude products was carried out by RP-HPLC using KNAUER 2501 HPLC system (Bad Homburg, Germany) and Phenomenex Luna (Torrance, CA, USA) C18 column (250 × 21.2 mm I.D.) with 10 µm silica (100 Å pore size). Experiments were carried out at a flow rate of 14 mL/min at room temperature. Linear gradient elution was applied. Gradient I: 0 min 5% B, 10 min 5% B, 10.1 min 20% B, 50 min 80% B. Gradient II: 0 min 20% B, 5 min 20% B, 50 min 80% B. Eluent A was 0.1% TFA in distilled water and eluent B was 0.1% TFA in CH_3_CN-water (80:20, *v*/*v*). Peaks were detected at *λ* = 220 nm.

Analytical RP-HPLC was performed on a Waters Symmetry (WAT 045905) C18 column (150 × 4.6 mm I.D.) with 5 µm silica (100 Å pore size) as a stationary phase. A linear gradient elution was developed: 0 min 0% B; 2 min 0% B; 22 min 90% B with eluent A (0.1% TFA in water) and eluent B (0.1% TFA in acetonitrile-water (80: 20, *v*/*v*)). A flow rate of 1 mL/min was used at ambient temperature. Samples were applied dissolved in eluent A and 20 μL was injected. Peaks were detected at *λ* = 220 nm.

### 2.5. Mass Spectrometry (MS)

The identification of the peptide analogues and conjugates was achieved by electrospray ionization mass spectrometry (ESI-MS) on a Bruker Daltonics Esquire 3000 Plus (Bremen, Germany) ion trap mass spectrometer, operating in continuous sample injection at 4 µL/min flow rate. Samples were dissolved in ACN-water (50:50 *v*/*v*%) mixture containing 0.1 *v*/*v*% AcOH. Mass spectra were recorded in positive ion mode in the *m/z* 50–2000 range.

For the stability and metabolism studies of the conjugates, liquid chromatography–mass spectrometry (LC-MS) analyses were performed on a Q ExactiveTM Focus, high resolution and high mass accuracy, hybrid quadrupole-orbitrap mass spectrometer (Thermo Fisher Scientific, Bremen, Germany) using on-line UHPLC coupling. UHPLC separation was performed on a Dionex 3000 UHPLC system using a Supelco Ascentis C18 column (2.1 × 150 mm, 3 µm). Linear gradient elution (0 min 2% B, 1 min 2% B, 17 min 90% B) with eluent A (0.1% HCOOH in water, *v*/*v*) and eluent B (0.1% HCOOH in acetonitrile/water, 80:20, *v*/*v*) was used at a flow rate of 0.2 mL/min at 40 °C. High-resolution mass spectra were acquired in the 200–1600 *m/z* range. LC-MS data were analyzed by XcaliburTM software (Thermo Fisher Scientific) and with Origin Pro 8 (OriginLab Corp., Northampton, MA, USA).

### 2.6. Measurement of Lysosomal Degradation of Conjugates by LC-MS

Conjugates were dissolved in distilled water in 2.5 μg/μL concentration followed by dilution with 0.2 M NaOAc solution (pH = 5.03) to 0.025 μg/μL. The lysosome-homogenate was prepared from rat liver and contained proteins in 16.6 μg/μL concentration. An aliquot (20 μL) of this stock solution was further diluted with 190 μL 0.2 M NaOAc solution, therefore the final protein concentration was 0.83 μg/μL. To prepare the reaction mixture, 15 μL (0.83 μg/μL) lysosome homogenate was added to 500 μL (0.025 μg/μL) conjugate solution. Furthermore, a control reaction mixture was always prepared which contained 500 μL conjugate solution and 15 μL NaOAc solution only. The solutions were stirred on 600 rpm at 37 °C and samples (50 μL) were taken out at 0 min, 5 min, 15 min, 30 min, 1 h, 2 h, 6 h, 24 h, and 72 h. The enzymatic activity was quenched by adding 5 μL formic acid to the samples. After this procedure, samples were frozen immediately at −25 °C. Control samples were taken at 0 min, 15 min, 1 h, 6 h, 24 h and 72 h. Composition of the samples was determined by HPLC-MS as described above.

### 2.7. Cell Cultures

For the in vitro characterization of conjugates four different tumor cell lines were used: PANC-1 (human pancreatic carcinoma of ductal origin), Colo-205 (human colorectal adenocarcinoma), A2058 (human metastatic melanoma) obtained from the European Collection of Authenticated Cell Cultures (ECACC, Salisbury, UK) and EBC-1 (human lung squamous cell carcinoma) purchased from the Japanese Research Resources Bank (Tokyo, Japan). Normal Human Dermal Fibroblasts (NHDF; Promocell, Heidelberg, Germany) as non-tumorous control cells were also investigated in order to determine the tumor selectivity of the proposed conjugates.

Dulbecco’s Modified Eagle Medium (DMEM, Lonza, Basel, Switzerland) was used for the culturing of the PANC-1, Colo-205 and EBC-1 cell lines, while the A2058 cell line was maintained in RPMI 1640 (Lonza). These basal media were supplemented with 10% fetal bovine serum (FBS, Gibco^®^/Invitrogen Corporation, New York, NY, USA), L-glutamine (2 mmol/L) (Lonza) and 100 µg/mL penicillin/streptomycin (Gibco^®^/Invitrogen Corporation). The medium of the Colo-205 cell line also contained 4500 mg/L D-glucose (Sigma-Aldrich, St. Louis, MO, USA), while, in case of EBC-1 cells, 1% non-essential amino acids (NEAA, Gibco^®^/Invitrogen Corporation) and 1 mM sodium pyruvate (Sigma-Aldrich) were also added to the culturing medium. For the cultivation of NHDF cells, Promocell Fibroblast Growth Medium (Promocell, Heidelberg, Germany) was used after adding SupplementMix (supplements necessary for the optimal growth of human fibroblasts, Promocell, Heidelberg, Germany) and the aforementioned antibiotics. All cell lines were grown in a T25 culture flask (Sigma-Aldrich or Eppendorf AG, Hamburg, Germany) in an incubator providing an atmosphere of 37 °C and 5% CO_2_.

### 2.8. Measurement of the In Vitro Cytotoxicity of Conjugates

The PANC-1 model cell line exhibits adherent properties under laboratory conditions; therefore, the potential effects of novel antitumor conjugates on cell viability were measured by impedimetry allowing the real-time detection of cell adhesion. This measurement is based on the registration of electrical resistance (impedance, *Z*) in alternating current (AC) field. The living cells transplanted to the gold measuring electrodes are physically insulated by phospholipid bilayer that covers them. This instrumentally measurable property changes (decreases) in response to cellular cytotoxic agents. Our measurements were performed on xCELLigence single plate (ACEA Biosciences, San Diego, CA, USA) dedicated for impedimetric analysis of cellular samples at 37 °C and 5% CO_2_.

During the initial phase of the experiments—the baseline recording—a special 96-well cell culture plate, E-plate (ACEA Biosciences), equipped with measuring electrodes, was pretreated with freshly prepared cell culture medium (for 60 min; sampling frequency: 1 min). Subsequently, PANC-1 cells were plated at a cell density of 10^4^ cells/well. During the 24 h incubation, the cells evenly covered the electrodes at the bottom of the wells of the E-plate. The resulting confluent cell cultures were then treated with the test substances at the following final concentrations: 10^−6^, 10^−5^, 10^−4^ M. Total treatment duration was 72 h and the sampling rate was 1 min (0–24 h); and then 15 min (48–72 h). In our measurements, three replicates were used, the control was the drug-free medium. The device displays the impedance change in the form of a cell index (CI), which is a relative (to the start of the experiment) and dimensionless index. The CI results were analyzed with xCELLigence RTCA 2.0 software and Origin Pro 8.0 software. Normalized CI values, expressed as a percentage of control, were used to characterize the cell viability and hence the effect of conjugates.

EBC-1 and Colo-205 model cells have weak/negligible adherent properties, and A2058 cells could not produce constant cell index values. Therefore, a colorimetric assay (alamarBlue-assay) was chosen instead of the xCELLigence system to investigate the viability of these model cells treated with the conjugates. Due to the growth characteristics of NHDF cells, alamarBlue-assay was performed also on this cell line.

The protocol for the alamarBlue-assay was similar to the method which was published earlier [26], with some minor modifications. Briefly, the cell seeding occurred on 96-well cell culture plates (Sarstedt AG, Nümbrecht, Germany) at 10^4^ cells/well concentration. After a 24 h long culturing period, the treatment was carried out with the conjugates at 10^−4^, 10^−5^ and 10^−6^ M final concentrations for 24, 48 and 72 h. In the next steps, the alamarBlue reagent (0.15 mg/mL, Sigma-Aldrich) dissolved in phosphate-buffered saline (PBS; pH = 7.2), was added to the wells. After 6 h incubation with the reagent, the fluorescence intensity of the samples was obtained by an LS-50B Luminescence Spectrometer (Perkin Elmer Ltd., Buckinghamshire, UK) or a Fluoroskan^TM^ FL Microplate Fluorometer and Luminometer (Thermo Scientific, Waltham, MA, USA) by using the following settings: *λ*_ex_ = 560 nm and *λ*_em_ = 590 nm.

Three parallels were performed per treatment group. In the case of controls, an equivalent volume of cell culture media was added to the cell. Fluorescence intensities of the samples treated with various concentrations of conjugates were expressed as a percentage of the fluorescence of control.

### 2.9. Flow Cytometric Measurement of Cell Surface Binding and Internalization

Cell surface binding and internalization of the conjugates were performed by flow cytometry (FACS-Calibur, Becton Dickinson, San Jose, CA, USA) based on the detection of the fluorescence activity of Dau (*λ*_ex_ = 488 nm, *λ*_em_ = 585 nm) linked to the peptides. Studies of binding and uptake were performed on PANC-1 cells.

Cells were seeded (2.5 × 10^5^ cells/mL, 900 μL/well) on 12-well plates, 24 h prior to the treatment with conjugates and free Dau. To distinguish the cell surface binding and internalization of conjugates, the cells were treated with the conjugate solutions at a final concentration of 10^−5^ M at two temperatures (37 °C and 4 °C) in parallel. After the incubation period of 30 min, cells were washed with PBS and were removed from the plate using TrypLE (Thermo Fisher Scientific, Waltham, MA, USA) cell-dissociation reagent, thus avoiding cell surface protein degradation. To stop the enzymatic dissociation, 500 μL of fresh medium was added to the wells after 3–5 min and the cells were transferred to FACS-tubes. After the centrifugation of the cell suspension, the cell pellets were resuspended in PBS (400 μL/tube) and the samples were measured by a flow cytometer.

To determine the fluorescence intensity and to evaluate the results CellQuest Pro (Becton Dickinson) and Flowing2.5.1. (Turku Center of Biotechnology, Turku, Finland) software were used. The measurement was carried out twice with two parallels per treatment group. Samples containing cells treated with fresh cell culture medium at 37 °C and 4 °C were used as negative controls. The instrument determines the relative fluorescence intensity of Dau built in the conjugates as geometric mean channel (GeoMean) value.

GeoMean values of the 4 °C and 37 °C samples were corrected with GeoMean values representing the autofluorescence of negative control samples. The fluorescence intensity of cells treated at 4 °C is proportional to the amount of conjugates bound to the cell surface, whereas the fluorescence intensity of cells treated at 37 °C is composed of the signal of conjugates internalized by the cells and those bound to the cell surface, too. The fluorescence intensity specific for the amount of conjugates internalized by the cells was calculated by subtracting GeoMean values of the cells incubated at 4 °C from GeoMean values of the samples incubated at 37 °C.

### 2.10. Experimental Animals

The Balb/c mice and immunodeficient SCID mice used in these studies were kept as described previously [27] and cared for according to the “Guiding Principles for the Care and Use of Animals” based upon the Helsinki Declaration, and they were approved by the local ethical committee. The permission license for breeding and performing experiments with laboratory animals: PEI/001/1738-3/2015 and PEI/001/2574-6/2015.

### 2.11. Acute and Chronic Toxicity Studies

Prior to the determination of in vivo antitumor activity, the acute and chronic toxicity studies of conjugate **4** (*Dau=Aoa*-GFLG-K(*Dau=Aoa*)-SKAAKN-*OH*) were investigated. Healthy Balb/c male mice (3 animals in each group with 29–33 g body weight) were used for these experiments. The conjugate was dissolved in sterile water for injection (Pharmamagist Kft., Budapest, Hungary) and injected in a volume of 0.1 mL/10 g body weight using the appropriate concentrations. In acute toxicity study, intraperitoneal (*i.p.*) administration of the conjugate **4** was carried out in 4 different doses: 3.125, 6.25, 12.5 and 25 mg/kg Dau-content. In chronic toxicity study, mice were treated with a dose of 10 mg/kg Dau-content of the conjugate on days 1, 3, 7, 9 and 11 (5 treatments). The toxicity was evaluated on the basis of life span, behavior and looking of the mice, as well as body weight. These parameters were followed for 14 days.

### 2.12. Mouse Model of Subcutaneous Human Pancreatic Cancer, Doses of Treatments and Measurements

For establishing pancreatic tumor in experimental animals, PANC-1 cells were used, which were cultured in RPMI 1640 medium (Lonza), supplemented with 10% heat-inactivated FBS (Biosera, Nuaille, France), and 1% penicillin/streptomycin (Lonza). They were cultured in sterile T175 flasks (Sarstedt AG) with a ventilation cap at 37 °C in a humidified atmosphere with 5% CO_2_.

Pancreatic cancer (PANC-1) cells were injected into SCID male mice (22–34 g) subcutaneously (*s.c.*), 3 × 10^6^ cells per animal in 200 µL M199 (Sigma-Aldrich) per animal. The (*i.p.*) administration of treatment started 10 days after cells inoculation when the average tumor volume was 36 mm^3^. Four groups with 7 animals per group were established and treated. The doses and schedule were as follows: control group was treated with sterile water used for the solubilization of Dau and the conjugates, while animals in the group administered with free Dau were treated on days 10, 19 and 24 after cell inoculation with a dose of 1 mg/kg. Mice administered with the conjugate **4** were separated in two groups and treated either with a dose of 10 mg/kg Dau-content (21.6 mg/kg conjugate) or with a dose of 2 mg/kg Dau-content (4.3 mg/kg conjugate) on days 10, 13, 19, 21, 24, 28, 31, 34, 39, 42, 46, 49, 53, 56, 60, 63, 67 and 70 after cells inoculation. Animal weight and tumor volumes were measured initially when the treatment started and at periodic intervals according to the treatment schedule. A digital caliper was used to measure the longest (a) and the shortest diameter (b) of a given tumor. The tumor volume was calculated using the formula *V* = ab^2^ × π/6, where a and b represent the measured parameters (length and width). The experiment was terminated on day 74 after cells inoculation (day 65 of treatment). Animals treated with free Dau had to be terminated after 3 treatments on day 28 after cells inoculation (day 19 of treatment) due to significant weight-loss. Animals were sacrificed by cervical dislocation; primary tumors and livers were harvested and weighted.

### 2.13. Statistical Analysis

For data on the cell viability assay, a one-way ANOVA algorithm of OriginPro 8.0 (OriginLab Corporation, Northampton, MA, USA) was used to assess the significance and calculate *p*-values. To compare the difference for all means, a Tukey’s post-hoc test was performed. In the case of the in vivo studies, statistical analyses were performed by GraphPad Prism 6 (GraphPad Software, San Diego, CA, USA) using the non-parametric Mann–Whitney test, where *p*-values lower or equal than 0.05 were considered statistically significant. The symbols *, **, and *** mean significant at *p* ≤ 0.05, *p* ≤ 0.01, and *p* ≤ 0.001, respectively.

## 3. Results and Discussion

### 3.1. Synthesis and Chemical Characterization of Conjugates

Five daunomycin-peptide conjugates were prepared for targeting PDAC cells. In our research, the potential homing peptide CKAAKN derived from phage display was modified by the replacement of Cys to Ser. The Cys/Ser exchange is commonly used in peptide chemistry, because of their similar structure (only thiol group is replaced by hydroxyl group), when there is no biological function of Cys. In CKAAKN the Cys was used for conjugation through its SH group (thioether linkage). This means that the free thiol group is not necessary for the biological activity; therefore, it can be replaced by other amino acids. Using another type of conjugation (e.g., oxime bond formation) the remaining free thiol group of Cys might cause unwanted disulfide bond formation that results in by-product. In addition, the Cys/Ser substitution can increase the hydrophilicity of the peptide that improves the water solubility of the peptide–drug conjugate.

Daunomycin (Dau) was applied as a payload. It was indicated in our previous research that Dau can be attached to peptides via oxime linkage easily with good yields. The oxime linked Dau-peptide conjugates showed circa one order of magnitude lower in vitro cytotoxic effect than the free drug. It might be because of the lower DNA binding affinity of the Dau containing metabolite compared to Dau [22]. However, the oxime linked conjugates are stable in the circulation and no loss of Dau can be observed before reaching the target cells. In addition, the conjugates are significantly better tolerated in vivo. Usually, they are not toxic up to 30 mg Dau content/kg body weight, while the free Dau can be applied at 1–2 mg/kg dose as maximum tolerated dose (MTD) [28]. Furthermore, the conjugates usually show similar or higher tumor growth inhibition at 10 mg Dau content/kg than the free drug at MTD with significantly less toxic side effects [27,29]. According to our observations the oxime linked Dau-peptide conjugates prevent better also the cell proliferation and metastases in mice models.

The syntheses of homing peptides were carried out by solid-phase peptide synthesis on Wang resin, that provides free carboxyl group on the *C*-terminus, using Fmoc/^t^Bu strategy. Prior to the cleavage of the peptides from resins, isopropylidene protected aminooxyacetic acid (> = *Aoa*-*OH*) was attached to the amino function(s) of peptides. The isopropylidene protecting group was cleaved from the purified peptide derivatives by 1.5 M methoxyamine in ammonium-acetate buffer at pH 5. The salt and the side products were removed by RP-HPLC. Purified peptide derivatives were linked via oxime bond to Dau overnight in ammonium-acetate buffer at pH 5 followed by an additional HPLC purification step. The synthesis route of a selected conjugate is presented on Scheme 1. Characteristics of the conjugates are given in the Appendix A with their metabolites obtained during lysosomal degradation.

### 3.2. In Vitro Cytotoxicity of Conjugates

The characterization of the antitumor effect of the prepared conjugates was performed primary on PANC-1 pancreatic ductal adenocarcinoma cells. This cell line was derived from a head pancreatic carcinoma with invasive phenotype and an ability to give metastasis to peripancreatic lymph node, thus PANC-1 cell line can be considered as an in vitro model of lymph-node-positive PDAC [23]. To obtain the selectivity of the conjugates, their cytotoxicity was also measured on Colo-205 colon adenocarcinoma, A2058 metastatic melanoma and EBC-1 lung squamous cell carcinoma cell lines. Due to the different adherent and growth characteristics of these model cells, two different methods were chosen for determining the antiproliferative/cytotoxic effect of the conjugates. Measurements on PANC-1 cells were conducted by an impedimetric technique (xCELLigence System) because of its ability to establish tight adhesion and large spread area. The viability of Colo-205, A2058 and EBC-1 cells were analyzed by a colorimetric assay (alamarBlue-assay). Colo-205 cells can grow in suspension and can partly attach to a tissue culture ware. Since the xCELLigence system monitors attached cells only, and any kind of effects (e.g., cytotoxic effect) causing alteration in cell adhesion or spreading, we expected that the cytotoxic effects of conjugates on Colo-205 cells would be therefore under-detected with impedimetry. EBC-1 cells are adherent, but they have a weak spreading capacity only. Thus, the attached membrane area, which determines fundamentally the usability of impedimetry [30], is in a small range. Therefore, the impedimetric measurement is not optimal or reliable for measuring the EBC-1 cell line. In addition, CI curves of A2058 cells demonstrated that these cells were not able to form a stable plateau phase, which would be required for the treatment and for reading the concentrations having an anti-tumor effect.

To obtain cytotoxicity using impedimetry, cells were treated for 24, 48 and 72 h by the 10^−6^, 10^−5^ and 10^−4^ M concentration solutions of the conjugates. The viability results of the 72 h treatment with conjugates at 10^−5^ M concentration are displayed in Table 1 as an example. The results show that the difference between the efficacy of the conjugates was well-observable in this concentration. The conjugates were ineffective at lower, 10^−6^ M concentration; when the conjugates were applied in a higher, 10^−4^ M concentration, most of the cells were killed after 72 h incubation.

The linear peptide conjugate **1** did not show any antitumor effect on PANC-1 cells (Table 1) up to 10^−5^ M concentration. The connection of *Dau=Aoa* to the *α*-amino group of extra Lys incorporated to the *N*-terminus of the peptide (conjugate **2**) and the presence of a Cathepsin B cleavable GFLG spacer [31] between the *Dau=Aoa* and the homing peptide (conjugate **3**) did not improve the activity of conjugate **1.** The modification with extra Lys was done to introduce an additional conjugation site (the *ε*-amino group of extra Lys), which provided the opportunity to create a branch with a second drug molecule.

The linear conjugate (**3**) with Gly-Phe-Leu-Gly (GFLG) spacer between the *Dau=Aoa* and the homing peptide was also used for the formation of conjugates with double Dau-content.

In contrast to the linear conjugates, the compounds with branched structure and two drug molecules decreased the cell viability significantly. Especially, conjugate **4**, in which *Dau=Aoa* was attached directly to the side chain of the Lys residue used for conjugation, presented high toxicity on PANC-1 cells. Less than 1% of the cells survived a 72 h treatment at a 10^−5^ M peptide concentration (Table 1). The incorporation of an extra GFLG spacer (conjugate **5**) into the branch decreased the potency of conjugate **4** (the cell viability was ca. 31% after 72 h). Nevertheless, a similar magnitude of cytotoxicity (viability: 34%) of conjugate **5** was already manifested after 24 h, while, in the case of conjugate **4**, only a slight antitumor activity (viability: 91%) was observed (Figure 1A).

It is important to note that the treatment with a conjugate could change the cell number and/or the morphology of the attached cells. In an impedimetric measurement, these changes could modify the CI and eventually the calculated viability parameter. In our previous study, the real-time curves showed a CI increase in the first 30 h of the treatment with a Dau-containing conjugate, but in the long term (40–72 h) the CI values were constantly decreased [32]. Depending on the efficacy of those conjugates, the transitional CI increase—probably as a result of morphological alterations—lasted shorter time interval and then turned into a decline of CI (cytotoxic effect) [32]. The treatment with conjugate **1** and **2** at 10^−5^ M for 72 h was not able to trigger cell death characterized by cell detachment and a decreased CI. However, our results still show that the treatment could irreversibly influence the cellular morphology (e.g., large and flat cell shape) which was manifested in increased CI values and cell viability higher than 100% (Table 1). This irreversible morphological alteration could evolve cell death by increasing incubation time and/or concentration. Real-time results of the highest concentration (10^−4^ M) of conjugate **1** and **2** support this theory. They caused an initial (0–36 h) increase in the CI values at 10^−4^ M concentration (viability after 24 h treatment: conjugate **1**: 224.1 ± 18.73%; conjugate **2**: 163.5 ± 24.3%) compared to the control. After this initial phase, the cell index values constantly decreased over the long term (after ~40 h), which represented their cytotoxic effect (viability after 24 h treatment; conjugate **1**: 17.1 ± 1.3% and conjugate **2**: 34.6 ± 9.4%).

To confirm that the different methods can be used to compare the effects of conjugates on the different model cells, the cytotoxicity of Dau was measured by both impedimetric and colorimetric methods on PANC-1 cells. In these experiments, very similar IC_50_ values—the concentration needed to decrease the cell viability by 50% (impedimetry: 1.89 × 10^−7^ M and colorimetry: 1.79 × 10^−7^ M)—were detected, which proved the suitability of these methods to evaluate the selectivity of the tested conjugates. Comparing the results of the viability measurements on different tumor cell lines, the conjugates **4** and **5** containing two drug molecules were proved to be more effective on PANC-1 cells than on melanoma (A2058), colon carcinoma (Colo-205) and lung carcinoma (EBC-1) cell lines. On the contrary, the linear conjugates (conjugates **1**, **2** and **3**), which were found to have no antitumor activity on PANC-1 cells, could exert a cytotoxic effect on the other three cell lines, especially on Colo-205 colon carcinoma cells (Appendix A).

Regarding the applicability of a SMDC for targeted tumor therapy it is important to investigate whether it has no or negligible effect on healthy cells. The conjugate **4**, as the most effective conjugate on PANC-1 cells, was selected for such an experiment. The effect of conjugate **4** on NHDF cell viability is depicted in Figure 1B, in comparison with its antitumor effect on PANC-1 cells (Figure 1A). The conjugate elicited a slight cytotoxic effect on NHDF cells, but this activity proved to be much lower than in case of PANC-1 cells. After 72 h incubation with 10^−5^ M conjugate **4**, roughly 75% of NHDF cells remained viable, while practically no viable PANC-1 cells were detectable. In the highest concentration (10^−4^ M), where most of the PANC-1 were killed already after 24 h, only 20% of NHDF cells were killed after 24 h. It seemed that the longer incubation time with conjugate **4** could not make cytotoxic effect be stronger in NHDF cells, since its time-dependent effect reached a plateau in a shorter incubation time (Figure 1B). Similar to the characteristics of the effects of conjugate **1** and **2** discussed above, the treatment with 10^−6^ M conjugate **4** resulted in cell viability values higher than 100% for PANC-1 cells. In this case as well, these results could be also due to irreversible morphological changes (e.g., large and flat cell shape) induce by conjugate **4.** The cells (so called senescent cells [33]) with altered morphology can go through apoptosis by increasing exposure time or concentration, which is recognizable in Figure 1A.

### 3.3. Cellular Uptake Measurements

PANC-1 cells were chosen to investigate the binding and the cellular uptake of the conjugates since these conjugates were designed for targeting pancreatic tumor cells and the most significant difference between their cytotoxic activities was also observed on this cell type.

The highest cellular uptake was detected in the case of the most cytotoxic conjugate **4** (Figure 2, Appendix A). Interestingly, the binding affinity of conjugate **2** was higher compared to conjugate **4**, but its cellular uptake was lower, although different was not significant. This may be explained by the more positive character of conjugate **2** that can slightly decrease internalization. However, the difference in cellular uptake of conjugates cannot completely explain the significant difference in their biological activity. Nevertheless, some correlations between these characteristics of conjugates can be established. Conjugates **1** and **3** entered the cells less efficiently than conjugate **4**, the ability of which correlated well with the binding affinity and can explain their neutral effect on inhibition of cell viability. This negative correlation was especially significant in the case of conjugate **3** (Figure 2). The difference in the binding affinity and cellular uptake of conjugates **4** and **5** might explain their in vitro antitumor effect.

If we compared the intracellular fluorescence intensity of conjugates with that of free Dau, which served as a positive control, a two orders of magnitude difference could be observed (Appendix A). This is not surprising, because the conjugates are assumed to enter the cells by receptor-mediated endocytosis, which has a lower capacity compared to the passive diffusion of free Dau. It was also previously reported that conjugation of Dau to a targeting peptide can decrease its fluorescence intensity by 90% compared to the free molecule [22]. Active metabolites released during the intracellular degradation (see in 3.4. chapter) of these constructs, have higher fluorescence than the intact conjugates. However, 30 min incubation time used in our cellular uptake measurements is short and generate a negligible amount of metabolites only. Therefore, the fluorescence intensity originated from the intact conjugate only was detectable and evaluable in the cellular uptake measurement.

### 3.4. Characterization of Lysosomal Degradation of Conjugates by LC-MS

The antitumor activity of SMDCs depends on many factors, such as receptor binding, cellular uptake, stability, lysosomal degradation, the structure of the resulted metabolite and the localization of the metabolite in the cell. As was presented above, in some cases the cell viability inhibition and the cellular uptake of the conjugates correlated well, but in other cases not. Therefore, it was reasonable to study the lysosomal degradation process, which determines the formation of active metabolites and consequently the antitumor activity of SMDCs.

The conjugates were reacted with lysosome homogenate, which contains lysosomal enzymes such as various proteases, for example, Cathepsin B, that can be overproduced in cancer cells and significantly influence the degradation of the conjugates. Under this procedure the peptide bonds are cleaved by the enzymes in various positions, and thus degradation products can be identified in the reaction mixture. The identification of the degradation products was performed by the LC-MS technique. In this work, we focused on the presence and the amounts of Dau-containing metabolites, because these compounds are presumed to be responsible for the cytotoxicity of the conjugates (Appendix A).

The release of free Dau was not detected from conjugates, however, various Dau-containing metabolites bearing one or a few amino acids were formed. The degradation of conjugates **1** and **2** was very fast. After 30 min, the presence of only small fragments: *Dau=Aoa*-S-*OH* and *Dau=Aoa*-SK-*OH* (Figure 3) can be detected in the case of conjugate **1** (Appendix A). Because the smallest metabolite releases quite fast, the law antitumor effect can be explained either by the low cellular uptake of this metabolite or its lower binding affinity to DNA [22]. The metabolites with OH groups on the amino acids have lower binding affinity to DNA than *Dau=Aoa*-Gly-*OH* or *H*-Lys(*Dau=Aoa*)-*OH*. In addition, the metabolite has to diffuse out from the lysosomes that might be also prevented by its hydrophilic character.

The *Dau=Aoa*-K-*OH*, *Dau=Aoa*-KS-*OH* (Figure 4) and *Dau=Aoa*-KSK-*OH* metabolites were detected in the case of conjugate **2** (Appendix A). The results indicate that the release of the smallest metabolite was rather slow. The pK_i_ value of this fragment is in a basic range, that might hinder the diffusion out of the metabolite from the lysosomes as well. This observation explains the low cytotoxic effect of this conjugate—even its cellular uptake was high.

Enzymatic degradation of conjugates containing a Cathepsin B cleavable GFLG spacer (**3**, **4**, **5**) resulted in the formation of *Dau=Aoa*-GFL-*OH; Dau=Aoa*-GF-*OH* and the *Dau=Aoa*-G-*OH* compounds as final products of the degradation. *Dau=Aoa*-G-*OH* is the smallest Dau-containing metabolite, furthermore, this compound is known to have a cytotoxic effect (Appendix A) [22].

Lysosomal degradation of conjugate **3** was rapid and completed within 1 h. After 0, 5, 30 and 60 min incubation, large fragments were detected lacking two or four amino acids from the C-terminus (*Dau=Aoa*-GFLGSKAA-*OH*, *Dau=Aoa*-GFLGKSK-*OH*). The detected fragments indicated that in this case, the peptidyl dipeptidase activity is predominant over the cleavage of the enzyme labile spacer. After 30 min, the presence of the two smallest fragments *Dau=Aoa*-GF-*OH* and *Dau=Aoa*-G-*OH* can be already detected. The intensity of the latter product—possessing cytotoxic property [22]—increased with increasing incubation time (Figure 5). After 72 h, only these two smallest metabolites were detected in the reaction mixture. However, the dipeptide containing fragment was still the main product.

After 72 h, the degradation of compound **4** resulted in similar products (*Dau=Aoa*-GFL-*OH*, *Dau=Aoa*-GF-*OH* and *Dau=Aoa*-G-*OH*) as the conjugate **3**. These metabolites originated from the cleavage of the GFLG spacer. Furthermore, additional Dau-containing metabolites could be identified in the case of conjugate **4**. After 2 h, the presence of the *H*-K (*Dau=Aoa*) SK-*OH* fragment could be detected as the main peak that was still the most intense peak after 72 h. In addition, the formation of *H*-K(*Dau=Aoa*)-*OH* (Appendix A) was also observed. This compound was also shown to have DNA binding affinity similar to the *Dau=Aoa*-G-*OH* [22]. The time-dependent increment in the amount of these two active metabolites could be observed (Figure 6). The *H*-K(*Dau=Aoa*)-*OH* metabolite with free *α*-amino and the carboxyl group of Lys has a rather neutral character because of the zwitterionic structure that might help the diffusion out of the lysosomes in contrast to *Dau=Aoa*-Lys-*OH*. Altogether, conjugate **4** showed the highest cellular uptake and efficient release of two active Dau-containing metabolites that correlate its highest in vitro cytotoxic effect.

The degradation of conjugate **5** in which both Dau are attached through GFLG spacer to the peptide backbone provided only the *Dau=Aoa*-GFL-*OH, Dau=Aoa*-GF-*OH* and *Dau=Aoa*-G-*OH* (Appendix A) fragments that increased in time. However, the degradation of conjugate **5** was not yet complete after 72 h. In this case, a larger metabolite (*H*-K(*Dau=Aoa*-GFLG) SK-*OH* (Appendix A) was still observed. The presence of this large molecular weight metabolite mentioned above suggested that the amount of the released *Dau=Aoa*-G-*OH* is derived rather from the *N*-terminus of the linear part, but its release is quite slow compared to the other conjugates (Figure 7). In comparison, compound **3** (*Dau=Aoa*-GFLGKSKAAKN-*OH*) was degradable by lysosomal enzymes quite fast resulting in the same metabolites, but the in vitro cytotoxic effect was much lower. This can be clearly explained by the very low binding efficiency and cellular uptake of conjugate **3**. The difference in the metabolite release between compound **3** and **5** suggests that the branching cause steric hindrance for enzyme degradation. It seems that this effect is higher when the flexible GFLG spacer is in the branch (compare with conjugate **4**).

### 3.5. Acute and Chronic Toxicity Studies of Conjugate 4

The conjugate **4** (*Dau=Aoa*-GFLGK(*Dau=Aoa*)SKAAKN-*OH*), that was the most efficient one in vitro experiments, was selected for in vivo studies. Prior to the treatment of tumor-bearing mice, acute (single *i.p.* administration) and chronic toxicity (5 *i.p.* treatments on days 1, 3, 7, 9 and 11) studies were conducted on healthy Balb/c male mice. These experiments were continued for 14 days, and no significant changes in the body weight, the general looking and the behavior of the animals, were observed during this period of time (Figure 8). Based on the results, it was concluded that the conjugate was not toxic at the applied concentrations and can be further investigated for its in vivo antitumor activity.

### 3.6. In Vivo Tumor Growth Inhibition of the Conjugate 4 in Pancreatic Tumor-bearing Mice

The tumor growth inhibition effect of conjugate **4** and free Dau on PANC-1 tumor-bearing SCID female mice was investigated. The free drug was applied once a week in the maximum tolerated dose of 1 mg/kg body weight while the conjugate was administered in average three times per week either in 2 mg/kg or in 10 mg/kg dose calculated for the Dau-content.

Body weight of *s.c*. human pancreatic cancer-bearing mice were evaluated. In comparison to the start of the experiment, the animal weight decreased significantly for 21% in the free Dau-treated group, and this group was terminated on day 28 after cell inoculation (day 19 of treatment) (Figure 9A). However, 35.7% inhibition of tumor growth (according to the tumor volume) was measured in this group, while in case of the groups treated with conjugate **4** at doses of 2 and 10 mg/kg Dau-content, the tumor growth inhibition was 6.8 and 27.9%, respectively, at this time point.

During the whole experiment, the animal weight stayed stable in conjugate **4** treated groups, with 0.8% decreasing under a dose of 2 mg/kg and 4% decrease in 10 mg/kg group. Animal weight decreased by 8.5% in the control group in comparison to the start of the experiment.

Antitumor effect of conjugate **4** was evaluated by measuring tumor volume in each group. At the end of the experiment (day 74) dose of 2 mg/kg, Dau-content inhibited tumor volume by 14%, while the dose of 10 mg/kg Dau-content inhibited significantly the tumor volume by 32.2% in comparison to the control group (Figure 9B). Interestingly, on the day 70 (day 61 of treatment), the inhibitions were 17.5%, and 39.9%, respectively, while on day 67 (day 58 of treatment) the dose of 10 mg/kg Dau-content elicited the most significant inhibition (43%) in the tumor volume compared to the control group. The data suggested that the tumor developed resistance during such a long treatment schedule.

At the end of the experiment (day 74), the animals were sacrificed and the antitumor effect of conjugate **4** was also evaluated by measuring tumor weight in each group (Figure 9C). Results showed that conjugate **4** inhibited tumor growth significantly under both doses (2 mg/kg Dau-content: 27.3%, 10 mg/kg Dau-content: 30.4%) in comparison with the tumor weight in the control group. Tumor weight depends on the content of tumor mass, necrotic cells, vessels, and general density which can differ for two tumors of the same size (same volume). Nevertheless, the correlation between the inhibition of the tumor volume and weight could be obtained at the end of the treatment with the lower dose, but the calculated SD values resulted in the significant inhibition of tumor weight (*p* = 0.05), while it was not significant for tumor volume [34]. The results indicated that the treatment with the higher dose of the conjugate might provide a higher shrinkage of tumor that could be suitable for preoperative chemotherapy.

The examination of the liver weight was carried out at the end of the experiment to determine the liver toxicity—as an indicator of toxic side effects—of conjugate **4** in *s.c.* human pancreatic cancer-bearing mice (Figure 9D). In the group treated with free Dau (three times with 1 mg/kg dose), the average liver/body weight ratio was significantly decreased by 29.1% compared to the control group, as well as in comparison to the groups treated with conjugate **4** (18 times with 2 or 10 mg Dau-content/kg dose); even so, the treatment period with free Dau was 46 days shorter than in case of the conjugate **4**. Average liver weight in conjugate **4** treated groups was not significantly changed in comparison to the control group.

To summarize the in vivo experiments, the data indicated that conjugate **4** could inhibit the tumor growth of *s.c.* human pancreatic cancer (PANC-1)-bearing mice significantly without causing any toxic side effects. Even a low dose (2 mg/kg Dau-content) of conjugate **4** could inhibit the tumor growth significantly, without any toxicity for the animals.

## 4. Conclusions

In this study, our goal was to select a potential peptide–drug conjugate for efficient targeted tumor therapy against pancreatic cancer. We developed five peptide-based drug conjugates against pancreatic cancer cells (PANC-1). SKAAKN hexapeptide was applied as targeting moiety which differs only at the *N*-terminal amino acid of CKAAKN that was efficiently used for drug delivery in a former study. Because in our construct thiol group Cys was not used as a conjugation site, it was replaced by Ser.

Three linear conjugates with one Dau and two-branched variants with two conjugated Dau were developed. The cytotoxicity, binding and cellular uptake of the conjugates were studied on PANC-1 cells. In addition, the degradation of the conjugates in lysosome homogenate was investigated as well. It was indicated that the efficacy of these peptide–drug conjugates was influenced by various factors. Our results indicate that at the development of peptide–drug conjugates, not only the cellular uptake of the conjugates but also the effective release of the active metabolite should be taken into account. The release of active metabolites was shown to depend highly on the structure of the conjugate. The most efficient conjugate was *Dau=Aoa*-GFLGK(*Dau=Aoa*)SKAAKN-*OH* (conjugate **4**) in the in vitro experiments, and it was selected for further in vivo studies. In contrast to the free drug, conjugate **4** did not show any toxicity during the treatment but elicited significant >30% tumor growth inhibition in PANC-1 bearing mice. The results suggested that SKAAKN peptide-based drug delivery systems could be promising constructs alone or in combination for the treatment of pancreatic cancers.

According to the results, conjugate **4** is suitable for further preclinical in vivo studies on more appropriate mice models representing better the stroma and the environment of PDAC. However, the models investigating stromal–tumor interactions are either induced (genetically modified) allograft models or patient-derived tumor xenografts (PDX). Both are very complex test tools; the use of the latter is gradually increasing, but due to the housing of special animals, the license of their use and the low efficiency of the establishment of tumors, they are not currently widespread in basic research.

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
