# Peer review of "Phage Display-Based Homing Peptide-Daunomycin Conjugates for Selective Drug Targeting to PANC-1 Pancreatic Cancer"

_pharmaceutics, 2020, doi:10.3390/pharmaceutics12060576_

Round 1

Reviewer 1 Report

In this work, Dókus and co-workers developed peptide-daunomycin conjugates for selective drug targeting to pancreatic ductal adenocarcinoma. This is an important topic of research given the high mortality rate of this type of cancer. This study also presents relevant results and appears to be made with care. Nevertheless, I still have some questions/comments that need to be addressed before a recomendation for acceptance can be given. Namely:

-It is not particularly well explained by the authors why substituting Cys by Ser was needed;

-The authors should discuss if changing Cys by Ser provided the peptides with new/different characteristics/properties;

-According to the present results, there is no release of free Dau from the peptides, but just Dau-based metabolites. Given this, it is possible that cytotoxic properties of these metabolites are different than free Dau?

-To further confirm the tumor selectivity of the proposed conjugates, their cytotoxicity should be tested in non-cancer cells;

Reviewer 2 Report

To the authors

Levente E. Dókus and colleagues report in this manuscript about Peptide-based drug for PDAC claiming a therapeutic role for homing peptide - daunomycin conjugates in PDAC. The manuscript covers an interesting topic, nonetheless, there are few sections that deserve to be restructured, in order to achieve the level and comprehensive overview that a journal like Pharmaceutics would aim to.

Major points to consider in subsequent versions:

Introduction the authors corrected mentioned the characteristics alluding to cancer malignant phenotype and its sensitiveness to both conventional and novel therapeutic agents. Nonetheless, it is well known that the stroma and the environment play a major role in PDAC (PMID: 30866547) and milieu reprogramming or interference with the tumour bystander interactions are of key importance for pancreatic ductal adenocarcinoma (PDAC) progression.

Discussion in the frame of this thinking, by discussing the paper from the PDAC non-tumoral niche would also highly important to mention the particular studies referring to the immune-microenvironment shaping and the related mechanisms within this particular PDAC model, with a particular interest to nodal diffusion. Indeed, a tight correlation exists between impaired immune-infiltrate, angiogenesis, stroma and cancer progression and dissemination to distant sites and to the nodal compartment. Indeed, stromal cells (namely myeloid and fibroblasts) and immune cells come and go across the permeable capillaries. Because of these intimate interactions, the capacity of PDAC cells and non-malignant counterpart can also be also discussed, since several examples have been recently published (i.e. PMID: 30866547 PMID: 25127858). Unfortunately, the author used a SCID mice model, characterized by the lack of a human tumour microenvironment, and the use of tumour cell-lines instead of fresh tumour biopsy tissues to establish the xenograft. This reviewer acknowledges that the establishment of an alternative in vivo model might be beyond the scope of this manuscript and maybe unrealistic due to the time constraints of spreading a novel finding. Nonetheless, these points should be discussed at least in terms of the strengths and weakness of the presented manuscript.

Finally, other groups uncovered the gene expression of locally advanced nodal positive disease to significantly impact in terms of pre-clinical translational impact in the past. Notably, PANC-1 cells have been reported to mimic a particularly nodal invasive prone PDAC model (PDAC- 31277479). Such cell lines model depicted an enrichment indicates that the proteins are at least partially biologically connected since WNT/CTNNB1 on cancer cells has shown to drive nodal invasive behaviour in N+ over N0 tumours (PMID: 31277479). Wrapping these pieces of evidence up, it is tempting to speculate that the wet cell models might represent a biological phenotype of the connection between PDAC invasiveness and immune-escape. In other words, conjugate 4, and daunomycin resistance/sensitivity can be influenced β-catenin/wnt signaling can, in turn, suppress chemokine production from tumour cells and prevents T cell infiltration and tumour aggressiveness, mimicking PANC-1 model (PMID: PMID: 25498972; PMID: 31277479).

Statistics: the authors state that they employed the Mann-Whitney test. This is fine as long as the data analyzed respect do a gaussian distribution. If this is the case, this should be clearly stated. Otherwise, the parametric test should be always performed.

General comments

Due to practical and ethical concerns associated with human experimentation, animal models have been essential in cancer research. However, the average rate of successful translation from animal models to clinical cancer trials is less than 10%. Animal models are limited in their ability to mimic the extremely complex process of human carcinogenesis, physiology and progression. Therefore, the safety and efficacy identified in animal studies are generally not translated into human trials. Animal models can serve as an important source of in vivo information, but alternative translational approaches have emerged that may eventually replace the link between in vitro studies and clinical applications.

Can the author comment on this and add their perspective on personal strategies prompted by their data in order to circumvent this limitation?

Minor

More generally, proofreading may further improve the quality of this well-written manuscript and enhanced image quality can be provided to better click the presented functional imaging results by high-resolution refinement.

Round 2

Reviewer 1 Report

The authors have addressed all my comments, and so, my recommendation is for acceptance.

Reviewer 2 Report

The authors have clarified several of the questions I raised in my previous review. Most of the major problems have been addressed by this revision.

No further comments from this reviewer.